Subject Area:
molecular biology/systems biology

Keywords:
CRISPR, ATR, cancer therapy, DNA replication, DNA repair

Author for correspondence:
Daniel Durocher
e-mail: durocher@lunenfeld.ca

†Present address: Ridgeline Therapeutics, Hochbergerstrasse 60C, CH-4057 Basel, Switzerland.
‡Present address: Netherlands Cancer Institute, Plesmanlaan 121, 1066 CX Amsterdam, The Netherlands.

# A consensus set of genetic vulnerabilities to ATR inhibition

Nicole Hustedt[1,†], Alejandro Álvarez-Quilón[1], Andrea McEwan[1], Jing Yi Yuan[1], Tiffany Cho[1,3], Lisa Koob[1,‡], Traver Hart[2] and Daniel Durocher[1,3]

[1]Lunenfeld-Tanenbaum Research Institute, Mount Sinai Hospital, 600 University Avenue, Toronto, Ontario, Canada M5G 1X5
[2]Department of Bioinformatics and Computational Biology, University of Texas MD Anderson Cancer Center, Houston, TX, USA
[3]Department of Molecular Genetics, University of Toronto, Toronto, Ontario, Canada M5S 1A8

DD, 0000-0003-3863-8635

The response to DNA replication stress in eukaryotes is under the control of the ataxia–telangiectasia and Rad3-related (ATR) kinase. ATR responds to single-stranded (ss) DNA to stabilize distressed DNA replication forks, modulate DNA replication firing and prevent cells with damaged DNA or incomplete DNA replication from entering into mitosis. Furthermore, inhibitors of ATR are currently in clinical development either as monotherapies or in combination with agents that perturb DNA replication. To gain a genetic view of the cellular pathways requiring ATR kinase function, we mapped genes whose mutation causes hypersensitivity to ATR inhibitors with genome-scale CRISPR/Cas9 screens. We delineate a consensus set of 117 genes enriched in DNA replication, DNA repair and cell cycle regulators that promote survival when ATR kinase activity is suppressed. We validate 14 genes from this set and report genes not previously described to modulate response to ATR inhibitors. In particular we found that the loss of the POLE3/POLE4 proteins, which are DNA polymerase ε accessory subunits, results in marked hypersensitivity to ATR inhibition. We anticipate that this 117-gene set will be useful for the identification of genes involved in the regulation of genome integrity and the characterization of new biological processes involving ATR, and may reveal biomarkers of ATR inhibitor response in the clinic.

## 1. Introduction

The ATR kinase is a phosphoinositide 3-kinase-like kinase that is activated when single-stranded (ss) DNA bound by the RPA complex is sensed by pathways anchored by the ATRIP or ETAA1 proteins [1–6]. Impaired replication or stalled replisomes often produce DNA structures that contain ssDNA that are then sensed by ATR [6–8]. Accordingly, ATR is a key modulator of DNA replication where it plays multiple roles in ensuring the orderly execution of DNA synthesis and its coordination with G2 phase entry [9,10]. Perhaps the best-characterized function of ATR is its role in controlling the timely activation of cyclin-dependent kinases (CDKs) via its activation of the CHK1 kinase [11,12]. The activated ATR-CHK1 pathway suppresses CDK activity by inactivating the phosphatases of the CDC25 family, which are CDK activators [13]. A second key role for ATR concerns its function in promoting the stability of distressed replication forks [14,15]. ATR impacts replication fork stability at multiple levels, for example by modulating fork reversal via the phosphorylation of proteins such as the annealing helicase SMARCAL1 [16], controlling the supply of dNTPs [17,18] and regulating the availability of RPA, which protects ssDNA from unscheduled nucleolysis [19]. ATR also controls DNA replication origin firing on both local and global scales [9,20].

royalsocietypublishing.org/journal/rsob Open Biol. 9: 190156

While ATR activation is known to suppress late origin firing in both unperturbed and challenged S phase [21–23], origins located in the vicinity of a blocked replication fork are shielded from this global inhibition, resulting in an increase in local origin firing that promotes the completion of DNA synthesis and the rescue of stalled replication forks [22,24–26].

These critical functions of ATR in coordinating the response to replication stress has made it an attractive therapeutic target in oncology given the observation that tumours often display signs of replication stress [27,28]. Multiple clinical-stage ATR inhibitors (ATRi) are now being tested in cancer treatment either as monotherapies or in combination with other agents [28,29]. While ATRi are showing single-agent efficacy in some patients, there is currently a paucity of robust biomarkers of responses, hampering development of this class of agents. Nevertheless, mutations in *ATM* or *ARID1A*, overexpression of *APOBEC3A/B*, as well as overexpression or rearrangements of the *EWSR1*, *MLL* and *SS18-SSX* genes have all been proposed as candidate patient-selection markers for ATRi clinical development [30–36].

We reasoned that the unbiased identification of genes promoting viability following ATR inhibition would be useful for two purposes. First, this list may contain genes that have not been previously associated with the regulation of DNA replication, cell cycle progression or DNA repair, and may reveal new facets of the function of ATR in promoting genome integrity. Second, this gene list may assist in the development of new patient-selection hypotheses or may reveal new genetic markers of ATRi response. Prior to the advent of CRISPR-based genetic screens, the search for genetic interactions with ATR deficiency involved studies in genetically tractable organisms like budding yeast or the use of RNA interference. For example, a focused screen for synthetic lethal interactions with a partially defective budding yeast ATR mutant, *mec1-100*, identified mutations in genes coding for chromatin remodellers, nuclear envelope components and various transcription regulators [37]. In another example, a focused siRNA screen in human cells surveying 240 DNA repair and replication genes identified deficiency of XPF-ERCC1 as well as knockdown of replication-related genes as conditions that induce ATRi sensitivity [38,39]. However, the advent of CRISPR/Cas9-based chemogenomic screens now allows the identification of vulnerabilities to ATR inhibition at the genome-scale level and in a robust manner, which was not previously possible with techniques such as RNA interference.

Therefore, we undertook four genome-scale CRISPR/Cas9 screens and combined their results with those of three additional, recently published screens [40]. From these seven screens, performed in five different cell lines and using two different ATRi, we describe a core set of 117 genes that promote cellular resistance to ATR inhibition. In particular, we found that loss-of-function mutations in the genes coding for the POLE3/POLE4 histone-fold complex cause ATRi hypersensitivity in human cells. POLE3 and POLE4 form an ancillary subunit complex of DNA polymerase ε involved in histone deposition at the replication fork [41,42]. Our results using a *POLE3* separation-of-function mutant suggest, however, that impaired histone deposition does not underlie the observed ATRi sensitivity, pointing rather to its function in DNA synthesis. We believe that this consensus genetic map of vulnerabilities to ATR inhibition will provide a useful resource to those interested in ATR function and therapeutics.

## 2. Results and discussion

To identify genes and cellular processes that require ATR kinase activity for cellular fitness, we undertook a set of four CRISPR/Cas9 somatic genetic screens in human cells. The screens, schematically depicted in figure 1*a*, were carried out essentially as described before [43,44]. They entailed the transduction of Cas9-expressing cells with a lentiviral library of single-guide (sg) RNAs, and after selection and time for editing, the resulting pool of gene-edited cells was split in two populations. One control population was left untreated for the duration of the screen while a second population was incubated with a sublethal dose of an ATR inhibitor that killed approximately 20% of cells. sgRNA abundance was determined in each population after 12 days of treatment by sequencing and a gene-based depletion score was determined with the most up-to-date version of drugZ [45].

We initially screened Flag-Cas9 expressing clones of three cell types: HCT116 cells, derived from a colon carcinoma; HeLa cells, derived from a cervical carcinoma; and a p53-mutated clone of RPE1 hTERT, which are telomerase-immortalized retinal pigment epithelial cells. These three cell lines were screened with the TKOv1 sgRNA library [44] using VE-821 as the ATR inhibitor [35]. In the fourth screen, the library used was the newer generation TKOv3 [46] and AZD6738 was employed as the ATR inhibitor [47]. The gene-level results can be found in electronic supplementary material, table S1.

Using a hit-selection threshold based on *p*-values < 0.001, we found 32, 34 and 130 genes that promoted ATRi resistance in the HCT116, HeLa and RPE1-hTERT $TP53^{-/-}$ cell lines, respectively, using the TKOv1/VE-821 combination (electronic supplementary material, table S1). In the RPE1-hTERT $TP53^{-/-}$ cell line screened with TKOv3 and AZD6728, 88 hits were found and there was a good agreement with the two RPE1 screens (figure 1*b*), with 41 common hits at *p* < 0.001 (electronic supplementary material, table S1). This good overlap suggests that both VE-821 and AZD6738 produce comparable phenotypes. In addition to these four screens, a recent publication also reported three CRISPR screens with AZD6738 as an ATR inhibitor in the MCF10A, HEK293 and HCT116 cell lines using the TKOv3 library [40]. We re-analysed this second set of screens using the newest version of drugZ [45] in order to provide a comparable set of data. We then combined the results of all seven screens and selected genes that were hits at a normalized *z*-score value (NormZ) less than −2.5 in at least two screens, which defined a set of high-confidence genes whose mutations cause ATR inhibitor sensitivity; this approach resulted in a 'core' set of 117 genes. Gene ontology (GO) enrichment analyses (figure 1*c,d*) indicated that this set is highly enriched in GO terms associated with DNA replication, DNA repair and DNA damage checkpoint such as Replication-born DSB repair via SCE (GO:1990414), DNA replication checkpoint (GO:0000076) and Recombinational repair (GO:0000725) among the top enriched terms. Similarly, GO term analysis for Cellular Component yielded Ribonuclease H2 complex (GO:0032299), the Fanconi Anaemia nuclear complex (GO:0043240) and site of double-strand break (GO:0035861) as the top 3 enriched terms. Analysis of

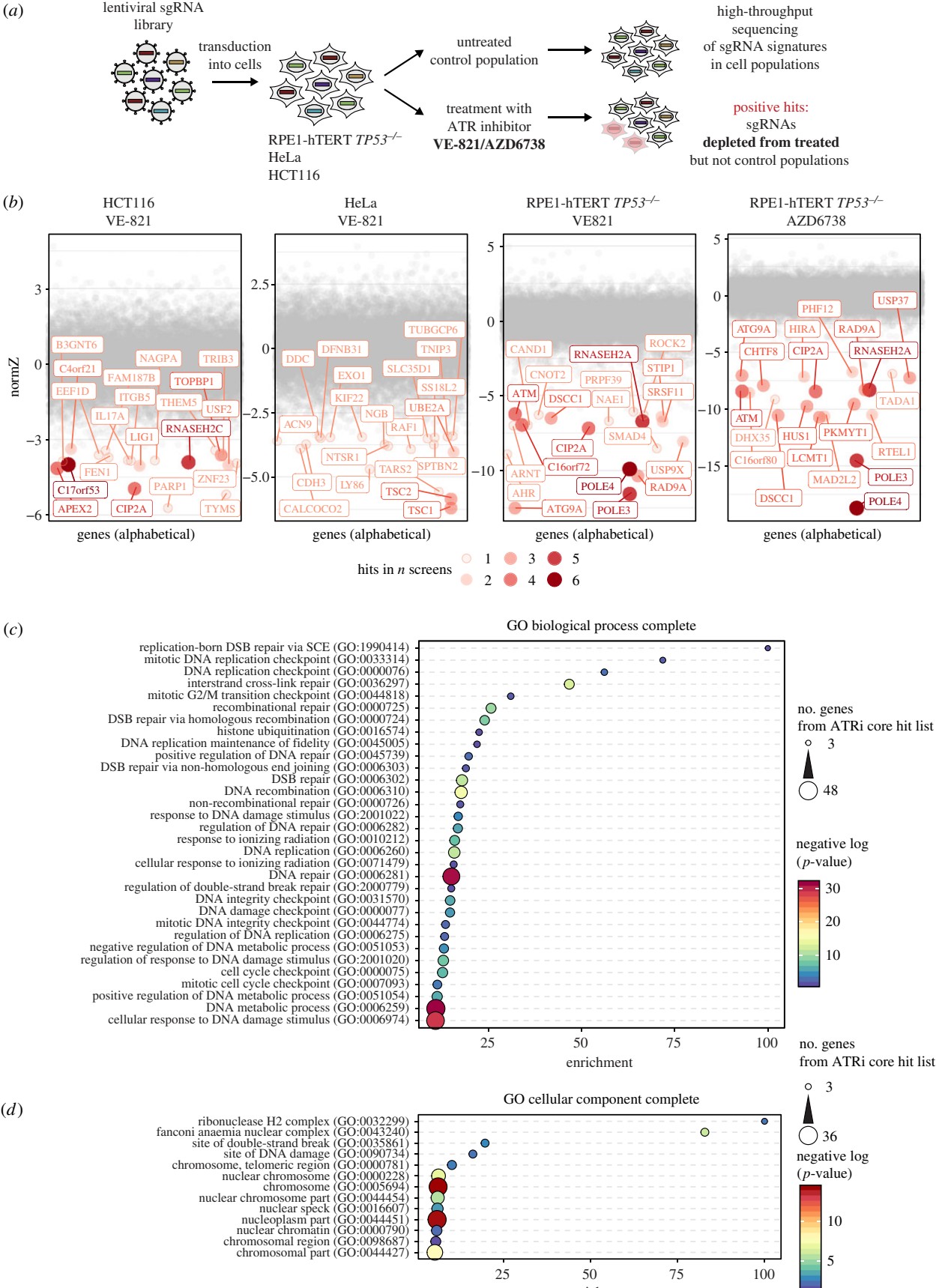

**Figure 1.** Identification of mutations that sensitize cells to ATR inhibition. (a) Schematic of genome-wide CRISPR/Cas9 screen work flow. (b) NormZ values were plotted against gene names in alphabetical order. For each screen, the genes with the 20 lowest NormZ values are labelled and coloured. Colour, size and transparency of circles indicate number of screens (our datasets and datasets from [40]) in which the genes were hits (i.e. showed NormZ values < −2.5). (c) Gene Ontology (GO) term enrichment analysis of Biological Process Complete terms (http://geneontology.org/page/go-enrichment-analysis) of 117 genes that were hits in at least two out of seven screens using default settings. Shown are GO terms that are enriched at least 10-fold. Circle size indicates number of genes from 117-gene core set included in each GO term, colour indicates negative log p-value and x-axis position indicates the fold enrichment compared to the whole genome reference set. (d) GO term enrichment analysis of Cellular Component Complete terms as in (c). Shown are GO terms that are enriched at least fivefold. DSB, DNA double strand break; SCE, sister chromatid exchange.

the gene set using the Reactome pathway database similarly identified highly connected pathways that revolve around DNA repair, DNA replication and cell cycle (electronic supplementary material, figure S1). A graphical comparison of the screens is shown in electronic supplementary material, figure S1c.

In this 117-gene core set, 11 genes were found as hits in at least four out of seven screens indicating that they are likely to modulate the response to ATR inhibition independently of cellular context. These genes were *APEX2*, *ATM*, *ATRIP*, *C16orf72*, *C17orf53*, *KIAA1524* (also known as *CIP2A*), *POLE3*, *POLE4*, *RNASEH2A*, *RNASEH2B* and *RNASEH2C*. The sensitization of ATM-deficient cells to ATR inhibition had been described before [35]. Similarly, ATRIP is a subunit of the ATR–ATRIP complex [6], and we (as well as the Cortez lab [38]) surmise that reducing ATR activity following *ATRIP* loss-of-function mutation sensitizes cells to ATR inhibition. In support of this possibility, ATR itself was a hit in RPE1-hTERT $TP53^{-/-}$ cells/AZD6738 ($p = 1.60 \times 10^{-7}$), in the HEK293/AZD6738 ($p = 0.0146$) and HeLa/VE-821 cells ($p = 0.0144$; electronic supplementary material, table S1). Drug sensitization by mutation of the drug target is a well-known phenomenon that has been harnessed to uncover drug targets in budding yeast [48]. The trimeric ribonuclease RNase H2 enzyme was recently described to promote resistance to ATR inhibitors [40]. RNase H2 also promotes resistance to PARP inhibition, and RNase H2-deficient cells experience replication-associated DNA damage that depends on topoisomerase I [44]. The replication-associated DNA lesions caused by defective ribonucleotide excision repair in RNase H2-deficient cells may cause this observed vulnerability to ATRi. *APEX2* codes for the APE2 nuclease, which has been implicated in the regulation of ATR activity in *Xenopus laevis* cell-free extracts [49] and was recently found to be synthetic lethal with BRCA1 and BRCA2 deficiency [50]. These findings support the notion that the 117-gene core set identifies genetic determinants of the response to ATR inhibition.

To functionally validate the results, we selected 18 genes that were hits in the screens carried out in our laboratory (electronic supplementary material, table S2). Of these, 15 out of 18 were part of the 117-gene core set. We undertook two-colour competitive growth assays in which Cas9-expressing cells were transduced with lentiviral vectors that simultaneously express an sgRNA that targets a gene of interest (*GOI*) as well as GFP, or a control virus that expresses an sgRNA targeting *LacZ* and mCherry (figure 2*a*). We carried out these assays first in RPE1-hTERT $TP53^{-/-}$ cells. Two independent sgRNAs were tested per gene, and in some cases we monitored indel formation by TIDE analysis [51] to ensure formation of loss-of-function mutations (electronic supplementary material, figure S2). The infected cell mixtures were grown in the presence or absence of VE-821 at doses of 2 and/or 4 μM, over the course of 15 days, and the proportions of GFP- and mCherry-expressing cells was determined at regular intervals by high-content microscopy. Out of this first set of analyses, the sgRNAs for 14 out of 18 genes clearly caused ATRi sensitivity (figure 2*b*), whereas the sgRNAs targeting the remaining four genes (*NAE1*, *DPH1*, *DTYMK* and *PPP1R8*) produced inconclusive results because the sgRNAs themselves were highly cytotoxic in the absence of ATRi treatment (electronic supplementary material, figure S3). However, as the set of 18 tested genes was not chosen at random, it is likely that the false positive

rate might be slightly higher than 22% (4/18). Remarkably, nearly all the genes surveyed that were part of the core set (13/15) were successfully validated. Importantly, we validated five (*APEX2*, *C17orf53*, *CIP2A*, *POLE3* and *POLE4*) of the 11 previously mentioned genes that were found in at least four out of seven screens, highlighting their importance in the response to loss of ATR function.

As a second stage of validation, we selected eight genes (*APEX2*, *C17orf53*, *CABIN1*, *CIP2A*, *DSCC1*, *POLE4*, *TOPBP1*, *TYMS*) and assessed the ability of sgRNAs targeting them to engender sensitivity both to a second ATRi (AZD6738) and a second cell line (HCT116 cells). We found that six out of eight genes promoted resistance to VE-821 and AZD6738 in both RPE1-hTERT $TP53^{-/-}$ and HCT116 cells (figure 3). The sgRNAs targeting *DSCC1* and *CABIN1* did not validate in HCT116 cells but we did not investigate further whether this was due to incomplete editing or whether it reflected biological differences between those cell lines.

As a final stage of validation, we generated clonal loss-of-function mutations in the *APEX2*, *CIP2A*, *POLE3* and *POLE4* genes (figure 4; electronic supplementary material, figures S4 and S5). We also added clones of *C16orf72* loss-of-function mutants as they were available in the laboratory (figure 4; electronic supplementary material, figure S4). We assessed sensitivity to AZD6738 in clonogenic survival assays and observed that disruption of each of these genes caused hypersensitivity to ATR inhibition, with the mutations in the *POLE3* and *POLE4* genes causing the greatest sensitization to ATR inhibition (figure 4*c*,*d*), in line with the results obtained in the competitive growth assay (figures 2 and 3).

The remarkable hypersensitivity of POLE3/POLE4-deficient cells to ATR inhibitors was intriguing in light of the recent characterization of this protein complex in chromatin maintenance and within the DNA Pol ε holoenzyme [41,42,52,53]. The POLE3/4 subunits form a histone-fold complex that is flexibly tethered to the core subunits of Pol ε but is not essential for DNA polymerization [53]. POLE3/4 acts as a histone H3-H4 chaperone that ensures symmetric histone deposition during DNA replication [41,42]. *Pole4*$^{-/-}$ mice are viable and show evidence of replication stress despite the fact that *POLE4*$^{-/-}$ cells have normal activation of the ATR pathway in response to a camptothecin or hydroxyurea challenge (figure 5*a*) [52]. We tested whether the ATRi sensitivity of *POLE3*$^{-/-}$ cells was due to defective histone deposition by complementing *POLE3*$^{-/-}$ knockout cells with a vector expressing a variant POLE3 with a C-terminal deletion, POLE3ΔC, which disrupts the histone deposition function of the POLE3/POLE4 dimer [41]. Expression of wild-type POLE3 or POLE3ΔC fully restored ATRi resistance, suggesting that histone deposition by this complex is unlikely to be involved in the normal cellular resistance to ATR inhibitors (figure 5*b*,*c*; electronic supplementary material, figure S6). Since siRNA-mediated depletion of Pol ε core subunits POLE leads to ATRi sensitivity [39], the ATRi sensitivity of POLE3/4-deficient cells may suggest that slight perturbances in Pol ε activity are sufficient to cause sensitivity to ATR inhibition. However, it is also possible that POLE3/POLE4 have additional roles at the replisome and that it is one of those activities that is responsible for the striking ATRi sensitivity of cells lacking these subunits.

In summary, our mapping of gene mutations that cause sensitivity to ATRi provides an unbiased view of the genetic architecture of the ATR-dependent control of genome integrity. We contend that this dataset will be a valuable tool for

royalsocietypublishing.org/journal/rsob Open Biol. 9: 190156

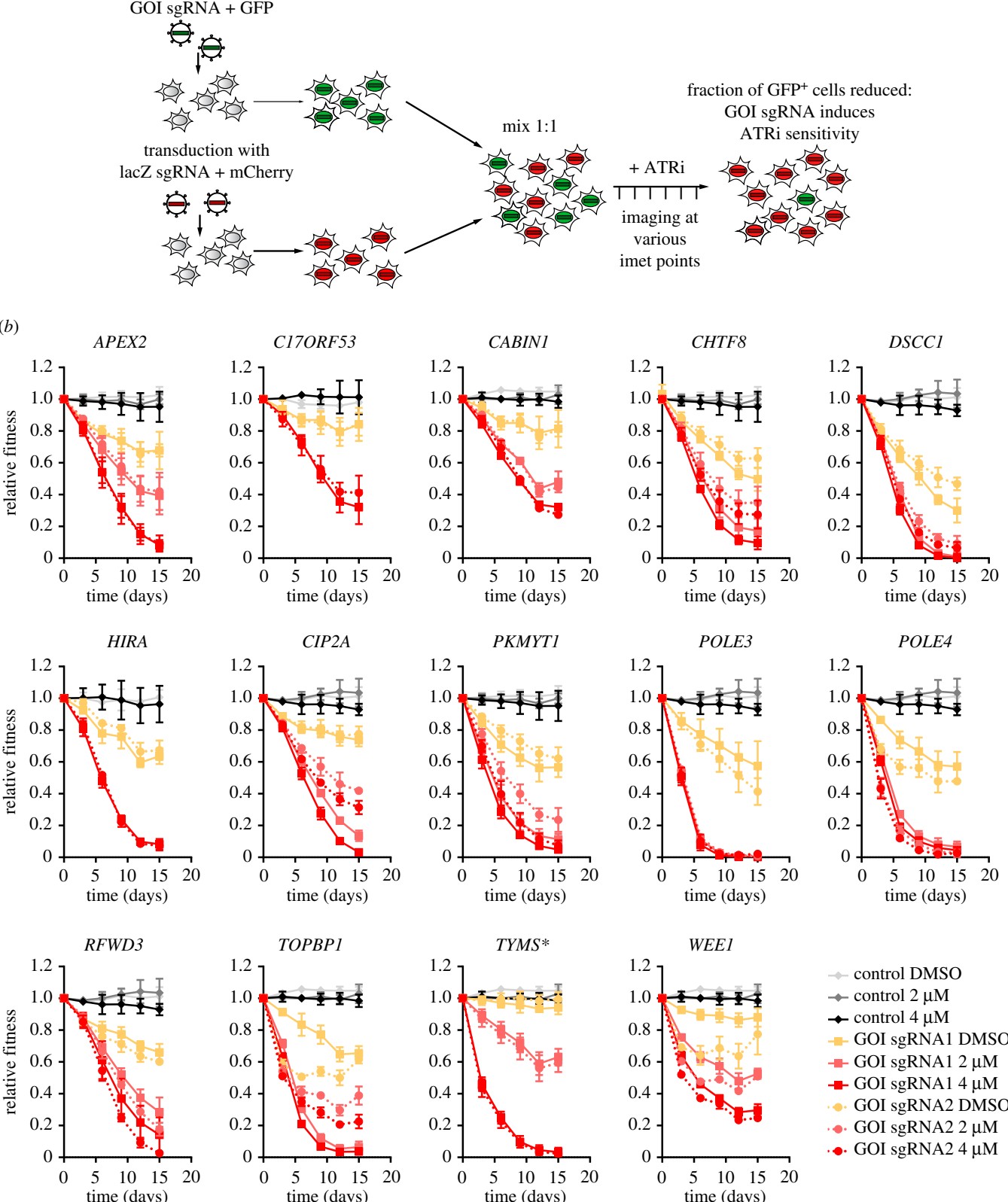

**Figure 2.** Hit validation using two sgRNAs for each gene of interest. (*a*) Schematic showing work flow of two-colour competitive growth assay. Cells were transduced with either an empty GFP vector (control) or vector with sgRNA targeting a gene of interest (GOI) coupled with GFP and an sgRNA targeting lacZ coupled with mCherry. GFP- and mCherry-expressing cells were mixed, treated or not with ATR inhibitor and population composition was followed over time. (*b*) Two-colour competitive growth assay results in RPE1-hTERT Flag-Cas9 TP53−/− cells using either empty GFP vector control or one of two sgRNAs targeting a GOI coupled with GFP as well as a sgRNA targeting lacZ coupled with mCherry. Populations were treated with indicated concentrations of ATR inhibitor VE-821 or vehicle (DMSO) throughout the duration of the experiment. HIRA- and C17orf53-targeted cells were treated with only 0 or 4 μM VE-821. Plotted are the fraction of GFP-positive cells normalized to T0. Asterisks indicate genes that are not part of ATRi core gene set. Error bars represent standard deviation of three biologically independent experiments.

royalsocietypublishing.org/journal/rsob Open Biol. 9: 190156

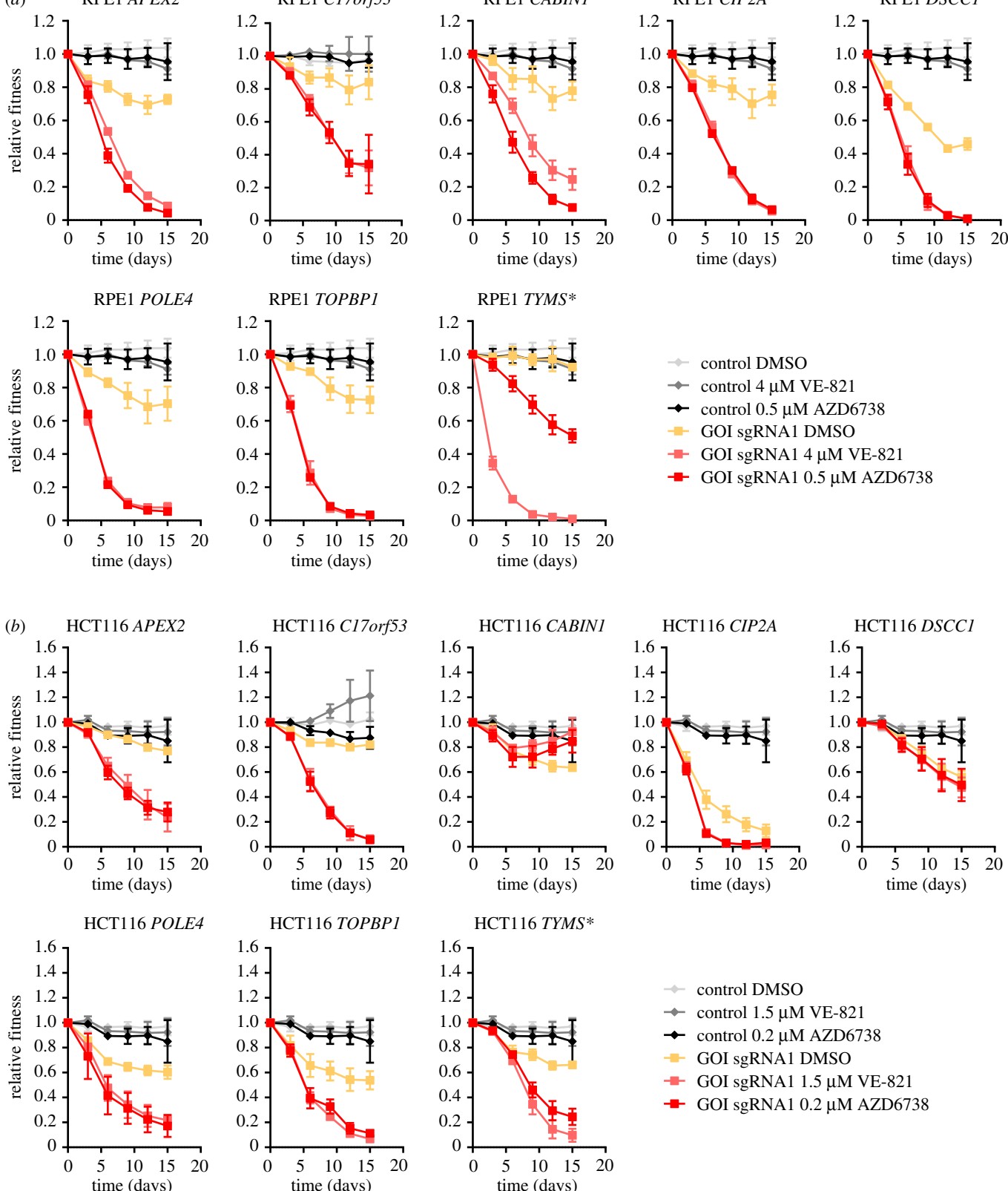

**Figure 3.** Hit validation using two ATR inhibitors and two cell lines. (*a*) Results from two-colour competitive growth assays using RPE1-hTERT Flag-Cas9 *TP53*$^{-/-}$ cells and the indicated concentrations of ATRi (VE-821 or AZD6738) or vehicle (DMSO). (*b*) Results from two-colour competitive growth assays as in panel (*a*), but using HCT116 Cas9 cells. Asterisks indicate genes that are not part of ATRi core gene set. Error bars represent standard deviation of three biologically independent experiments.

both clinical and biological researchers. Beyond revealing potential new biomarkers of the ATRi response, this list is rich in potential new avenues for study. For example, there are many genes in this list that have never been characterized in-depth for a role in genome integrity. A prime candidate is *C4orf21*, also known as *ZGRF1*, which encodes a protein containing a GRF-type zinc finger, a domain found in bona fide

DNA repair proteins such as TOPIIIα, APE2 and the NEIL3 glycosylase. In some other cases, the inclusion of a gene in this list cements the potential role of their encoded protein as a modulator of genome integrity. An example is DHX9, which is an RNA helicase that promotes R-loop formation and impedes DNA replication [54]. We also note that this core gene set is unlikely to be exhaustive because fitness

royalsocietypublishing.org/journal/rsob    Open Biol. 9: 190156

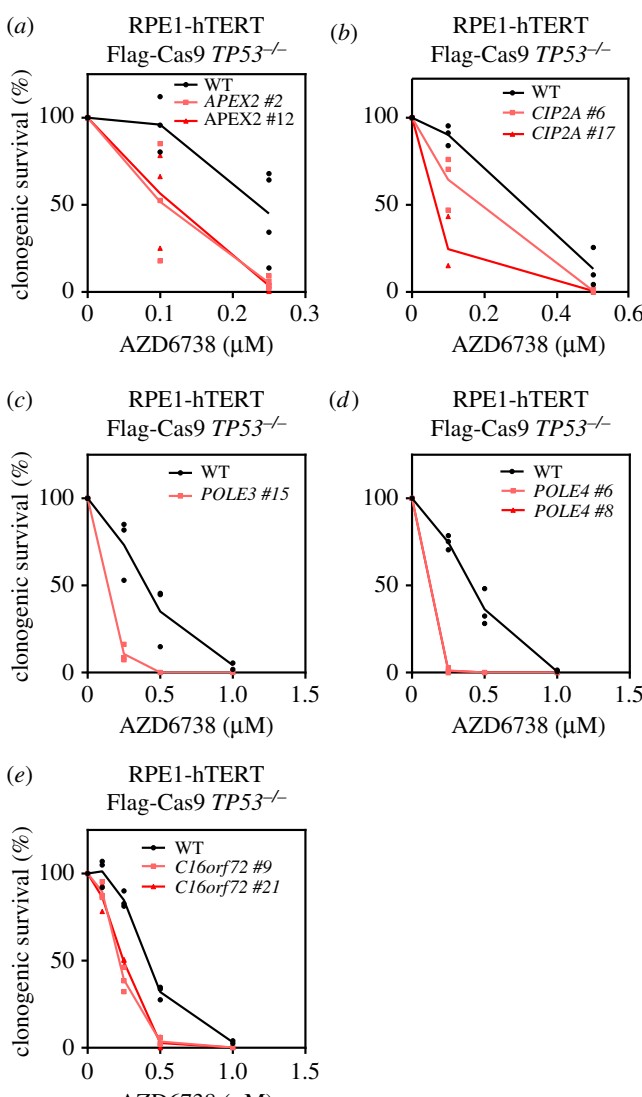

**Figure 4.** Clonal KO cell lines of APEX2, CIP2A, POLE3, POLE4 and C16ORF72 are sensitive to ATRi. (*a*) Clonogenic survival of RPE1-hTERT Flag-Cas9 TP53−/− (WT) and two RPE1-hTERT Flag-Cas9 *TP53*−/− *APEX2*−/− clones treated with indicated concentrations of ATR inhibitor AZD6738. (*b*) As in (*a*) using two *CIP2A*−/− clones. (*c*) As in (*a*) using a *POLE3*−/− clone. (*d*) As in (*a*) using two *POLE4*−/− clones. (*e*) As in (*a*) using two *C16orf72*−/− clones. Data are from three biologically independent experiments.

screens necessitate that the sgRNAs have some level of representation in the control cell population and therefore can miss cell-essential genes. Given that DNA replication is an essential process associated with the ATR pathway, it is likely that a comprehensive list of vulnerabilities to ATR inhibition will necessitate other approaches such as phenotypic screens or CRISPR interference. Finally, our finding that mutations in *POLE3/POLE4* cause hyper-sensitivity to ATRi is particularly intriguing and forms the basis of ongoing studies. Our data indicate that the sensitivity imparted by the loss of POLE3/POLE4 is not due to defective histone deposition, suggesting it is rather caused by a defect in DNA synthesis that remains to be uncovered.

# 3. Methods

## 3.1. Plasmids

DNA sequences corresponding to sgRNAs were cloned into LentiGuide-Puro (Addgene: 52963) or a modified form of LentiCRISPRv2 (Addgene: 52961) in which the sequence encoding Cas9 was replaced by that for NLS-tagged GFP or mCherry using AgeI and BamHI (designated as LentiGuide-NLS-GFP or –mCherry), as described [55,56]. GFP-POLE3 full length and GFP-POLE3 with amino acid residues 113–140 deleted (GFP-POLE3ΔC) were cloned between NheI and AgeI restriction sites of pCW57.1 (Addgene: 41393).

## 3.2. Cell lines and gene editing

293T cells were obtained from ATCC. HeLa Flag-Cas9 and RPE1-hTERT Flag-Cas9 *TP53*−/− were published earlier [44] and HCT116 Flag-Cas9 cells [43] were a kind gift from Jason Moffat. RPE1-hTERT Flag-Cas9 *TP53*−/− were grown in Dulbecco's Modified Eagle Medium (DMEM; Gibco/Wisent) supplemented with 10% fetal bovine serum (FBS; Wisent), 200 mM GlutaMAX, 1× non-essential amino acids (both Gibco) and 100 U ml−1 penicillin and 100 μg ml−1 streptomycin (Pen/Strep; Wisent/Gibco). HeLa Flag-Cas9 and 293T cells were cultured in DMEM supplemented with 10% FBS and Pen/Strep. HCT116 Flag-Cas9 cells were cultured in McCoy's 5A medium (Gibco) supplemented with 10% FBS and Pen/Strep. Cell lines stably expressing Flag-Cas9 were maintained in the presence of 2 μg ml−1 blasticidin.

Lentiviral particles were produced in 293T cells by co-transfection of the targeting vector with plasmids expressing VSV-G, RRE and REV using TransIT LT-1 transfection reagent (Mirus). Viral transductions were performed in the presence of 4–8 μg ml−1 polybrene (Sigma-Aldrich) at an MOI < 1. Transduced RPE1-hTERT Flag-Cas9 *TP53*−/− and HCT116 Cas9 cells were selected by culturing in the presence of 15–20 or 2 μg ml−1 puromycin, respectively.

*APEX2, CIP2A, POLE3 and POLE4* gene knockouts were generated in RPE1-hTERT Flag-Cas9 *TP53*−/− cells by electroporation of LentiGuide-Puro or LentiGuide-NLS-GFP vectors using an Amaxa II Nucleofector (Lonza). *C16orf72* gene knockout clones were generated in RPE1-hTERT Flag-Cas9 *TP53*−/− cells by transfecting sgRNA/Cas9 ribonucleoprotein complex using Lipofectamine CRISPRMAX Cas9 Transfection Reagent (Invitrogen). After 24 h transfection, cells were expanded, followed by single clone isolation. For sgRNA sequences employed see electronic supplementary material, table S3; *APEX2* #12: *APEX2*sgRNA1, *APEX2* #2: *APEX2*sgRNA2, *C16orf72* #9 and *C16orf72* #21: C16orf72sgRNA1, *CIP2A* #6 and *CIP2A* #17: CIP2Asg RNA2, *POLE3* #15: *POLE3*sgRNA1, *POLE4* #6 and *POLE4* #8: *POLE4*-sgRNA1. Twenty-four hours following transfection, cells were selected for 24–48 h with 15–20 μg ml−1 puromycin, followed by single clone isolation. Gene mutations were further confirmed by PCR amplification, DNA sequencing and TIDE analysis [51]. For primers used for genomic PCR, see electronic supplementary material, table S4. Loss of gene expression was further confirmed either by immunoblotting to assess protein levels if antibodies were available (see below) or by RT-qPCR to assess mRNA levels using GAPDH for normalization. Taqman assays employed were GAPDH (Hs99999905_m1) and APEX2 (Hs00205565_m1) from Thermo Fisher Scientific.

RPE1-hTERT Flag-Cas9 *TP53*−/− (WT) or *POLE3*−/− #15 cells expressing GFP, GFP-POLE3 or GFP-POLE3ΔC were generated by transduction with lentiviral particles of

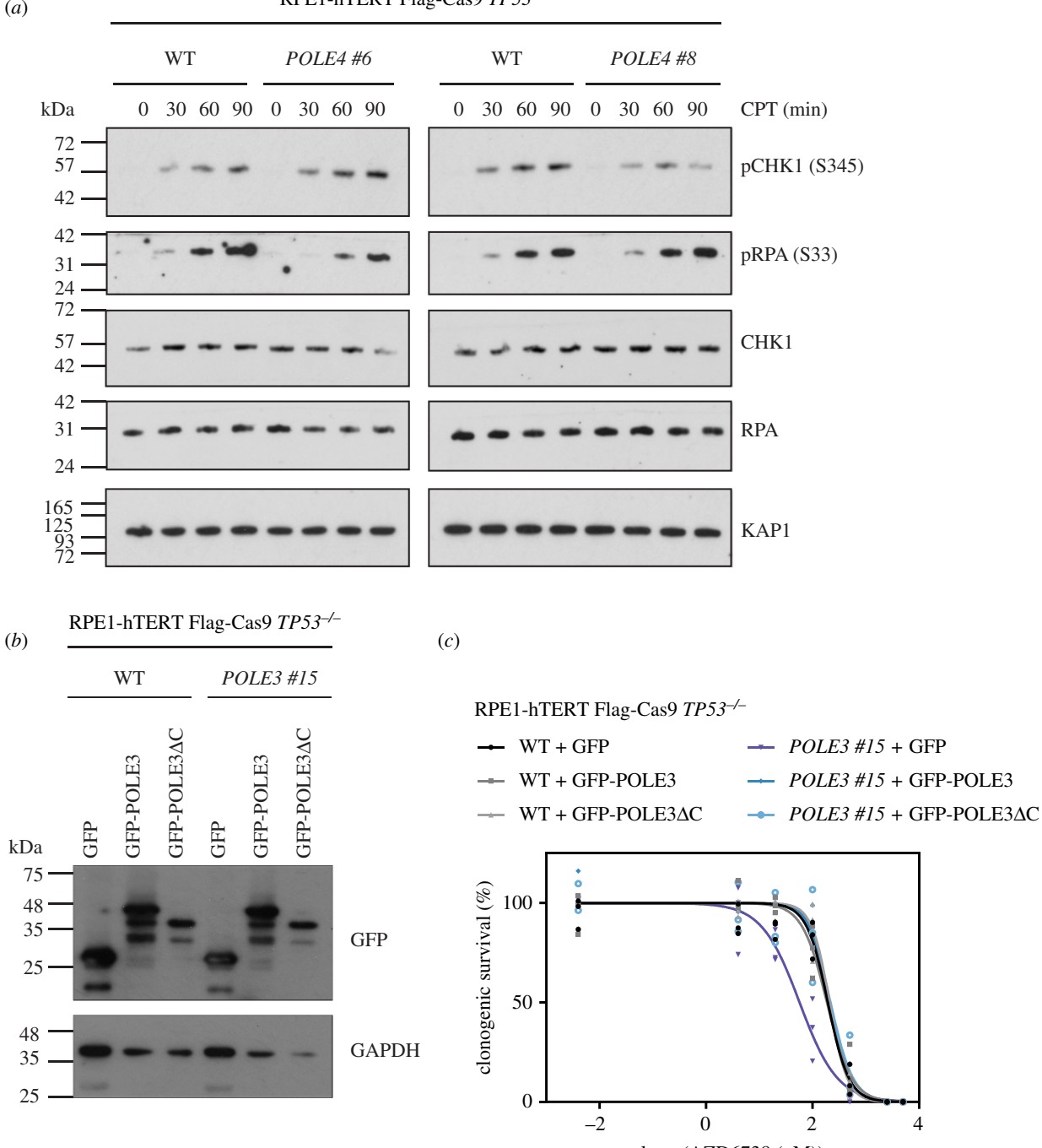

**Figure 5.** ATRi sensitivity in POLE3$^{-/-}$ or POLE4$^{-/-}$ cells is not caused by defective ATR signalling or histone deposition. (a) Whole cell extracts from wild-type RPE1-hTERT Flag-Cas9 TP53$^{-/-}$ (WT) or the indicated POLE4$^{-/-}$ clones treated with 1 µM camptothecin (CPT) were used for immunoblotting with indicated antibodies. pCHK1 and pRPA refer to phosphorylated proteins; brackets indicate modified amino acid residues. KAP1 served as loading control. (b) Whole cell extracts from WT or the indicated POLE3$^{-/-}$ clone expressing GFP, GFP-POLE3 or GFP-POLE3ΔC were used for immunoblotting with indicated antibodies. GAPDH served as loading control. (c) Clonogenic survival of WT or the indicated POLE3$^{-/-}$ clone expressing GFP, GFP-POLE3 or GFP-POLE3ΔC treated with indicated concentrations of ATR inhibitor AZD6738. Data are from three biologically independent experiments.

pCW57.1-derived GFP, -GFP-POLE3 or -GFP-POLE3ΔC constructs and subsequent selection with 20 µg ml$^{-1}$ Puromycin for 48 h. Cells were maintained in the presence of 5 µg ml$^{-1}$ puromycin and 1 µg ml$^{-1}$ doxycycline.

## 3.3. Antibodies, siRNAs and drugs

The following antibodies were used in this study at the indicated dilutions: anti-CIP2A (CST #14805; 1 : 1000), anti-

GAPDH (Sigma-Aldrich G9545; 1 : 20 000), anti-GFP (gift from Laurence Pelletier, 1 : 10 000), anti-KAP1 (Bethyl A300-274A, 1 : 5000), anti-POLE3 (Bethyl A301-245A-1; 1 : 2000), anti-POLE4 (Abcam ab220695; 1 : 200), anti-Tubulin (Millipore CP06, 1 : 2000), anti-pCHK1 (S345) (Cell Signaling #2348, 1 : 1000), anti-CHK1 (Santa Cruz sc8408, 1 : 500), anti-pRPA32 (S33) (Bethyl A300-246A-3, 1 : 20 000), anti-RPA32 (Abcam ab2175, 1 : 500). The following secondary antibodies for immunoblotting were used in this study:

royalsocietypublishing.org/journal/rsob   Open Biol. 9: 190156

peroxidase-conjugated AffiniPure Bovine Anti-Goat IgG (Jackson Immuno Research 805-035-180) and peroxidase-conjugated Sheep Anti Mouse IgG (GE Healthcare NA931 V). All peroxidase-conjugated secondary antibodies were used at a dilution of 1 : 5000. Protein bands were detected using the SuperSignal West Pico enhanced chemiluminescence reagent (Thermo Fisher Scientific). The following siRNAs from Dharmacon were used in this study: control, siGENOME Non-targeting Pool #2 (D-001206-14-05); POLE3, siGENOME SMARTpool (M-008460-01-0005); POLE4, siGENOME SMART-pool (M-009850-01-0005); APEX2, siGENOME SMARTpool (M-013730-00-0005). ATR inhibitors VE-821 and AZD6738 were purchased from SelleckChem.

### 3.4. CRISPR/Cas9 screens

RPE1-hTERT Flag-Cas9 $TP53^{-/-}$, HeLa Flag-Cas9 and HCT116 Flag-Cas9-expressing cells were transduced with the lentiviral TKOv1 library at a low MOI (approx. 0.35) and puromycin-containing medium was added the next day to select for transductants. Selection was continued until 72 h post-transduction, which was considered the initial time point, t0. To identify VE-821 sensitizers, the negative-selection screen was performed by subculturing at days 3 and 6 (t3 and t6), at which point the cultures were split into two populations. One was left untreated and to the other a dose of VE-821 amounting to 20% of the lethal dose ($LD_{20}$) (HeLa Flag-Cas9, 1.5 μM; HCT116 Flag-Cas9 1.5 μM; RPE1-hTERT Flag-Cas9 $TP53^{-/-}$, 4 μM) was added. Cells were grown with or without VE-821 until t18 and subcultured every 3 days. Sample cell pellets were frozen at t18 for genomic DNA (gDNA) isolation. Screens were performed in technical triplicates and a library coverage of greater than or equal to 200 cells per sgRNA was maintained at every step. The AZD6738 screen was performed at a concentration of 0.5 μM AZD6738 using the TKOv3 library [46] in technical duplicates and a library coverage of greater than or equal to 375 cells per sgRNA was maintained at every step. Genomic DNA from cell pellets was isolated using the QIAamp Blood Maxi Kit (Qiagen) and genome-integrated sgRNA sequences were amplified by PCR using the KAPA HiFi HotStart ReadyMix (Kapa Biosystems). i5 and i7 multiplexing barcodes were added in a second round of PCR and final gel-purified products were sequenced on Illumina NextSeq500 or HiSeq2500 systems to determine sgRNA representation in each sample. sgRNA sequence read counts (electronic supplementary material, table S5) were obtained using MAGeCK [57]. drugZ [45] was used to identify gene knockouts which were depleted from ATRi-treated t18 populations but not depleted from untreated cells.

### 3.5. Two-colour competitive growth assay

Cells were transduced with either virus particles of NLS-mCherry LacZ-sgRNA or NLS-GFP GOI-sgRNA. Twenty-four hours after transduction transduced cells were selected using 15–20 μg ml$^{-1}$ puromycin for 48 h. At this time mCherry- and GFP-expressing cells were mixed 1 : 1 (2500 cells + 2500 cells) and plated in a 12-well format. Cells were imaged for GFP- and mCherry signals 24 h after initial plating ($t = 0$) and ATR inhibitors were added subsequently

(RPE1-hTERT Flag-Cas9 $TP53^{-/-}$: 2 μM/4 μM VE-821 or 0.5 μM AZD6738; HCT116 Flag-Cas9: 1.5 μM VE-821 and 0.2 μM AZD6738). During the course of the experiment, cells were subcultured when near-confluency was reached and imaged on days 3, 6, 9, 12 and 15. An InCell Analyzer system (GE Healthcare Life Sciences) equipped with a 4× objective was used for imaging. Segmentation and counting of the number of GFP-positive and mCherry-positive cells was performed using an Acapella script (PerkinElmer). Efficiency of indel formation for a subset of sgRNAs was analysed by performing PCR amplification and sequencing of the region surrounding the sgRNA target sequence and TIDE analysis on DNA isolated from GFP-expressing cells 9 days post-transduction.

### 3.6. Clonogenic survival assays

RPE1-hTERT Flag-Cas9 $TP53^{-/-}$ cells were seeded in 10-cm dishes (WT: 500 or 1000 cells; $APEX2^{-/-}$: 1000–1500 cells; $CIP2A^{-/-}$: 2000 cells; $POLE3^{-/-}$: 1000 cells; $POLE4^{-/-}$: 1000 cells) in the presence AZD6738 or left untreated. For WT or $POLE3^{-/-}$ cells transduced with pCW57.1-GFP/GFP-POLE3/GFP-POLE3ΔC 1000 cells were seeded in 10 cm dishes in the presence of 1 μg ml$^{-1}$ doxycycline and in the presence or absence of AZD6738. Doxycycline-containing medium +/− AZD6738 was refreshed every 3 days. After 11–14 days, colonies were stained with crystal violet solution (0.4% (w/v) crystal violet, 20% methanol) and counted manually. Relative survival was calculated for the drug treatments by setting the number of colonies in non-treated controls at 100%.

### 3.7. Incucyte assay for ATRi sensitivity

RPE1-hTERT Flag-Cas9 $TP53^{-/-}$ (WT) or $POLE3^{-/-}$ cells were seeded into a 12-well plate (2500 cells per well) in the presence of 1 μg ml$^{-1}$ doxycycline. After 24 h medium containing 1 μg ml$^{-1}$ doxycycline and AZD6738 to the desired final concentration was added. Every six hours 25 images per well were acquired using an Incucyte S3 Live Cell Analysis System (Sartorius) and analysed for percentage confluency using Incucyte S3 2018A software (Sartorius). Doxycycline-containing medium +/− AZD6738 was refreshed after 3 days. Percentage confluency at the 138 h time point was used to calculate relative survival by setting the percentage confluency in non-treated controls at 100%.

Data accessibility. The gene-level normalized Z-scores (drugZ scores) and the sgRNA-level read counts are available as electronic supplementary material, tables S1 and table S5, respectively.

Competing interests. D.D. and T.H. are advisers to Repare Therapeutics.

Funding. N.H. is supported by a Human Frontier Science Program long-term fellowship. T.H. is supported by MD Anderson Cancer Center Support grant no. P30 CA016672 and the Cancer Prevention Research Institute of Texas (no. CPRIT/RR160032). D.D. is funded by CIHR grant no. FDN143343 and Canadian Cancer Society (CCS grant no. 705644).

Acknowledgements. We thank Rachel Szilard for the critical reading of this manuscript; Michal Zimmermann and Salomé Adam for their help during the CRISPR screens; Jason Moffat for TKO libraries and cell lines; Laurence Pelletier for sharing his GFP antibody; and Kin Chan of the LTRI NBCC for sequencing. D.D. is a Canada Research Chair (Tier I).

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
