## [Reviewer comments · Open Biology]

Review History

RSOB-19-0156.R0 (Original submission)

Review form: Reviewer 1

Recommendation

Accept with minor revision (please list in comments)

Do you have any ethical concerns with this paper?

No

Comments to the Author

In this manuscript the authors report the results of CRISPR-Cas9 screens looking for genetic vulnerabilities to ATR inhibition. The results provide a useful resource for investigators in the DNA repair, replication, and DNA damage signaling fields. The writing and figures are clear and the conclusions are appropriate. I recommend publication. The following comments may assist the authors in improving the final version of the manuscript.

1. It is difficult to judge overlap between the 4 screens in the three cell types. I recommend including a simple Venn diagram.

2. Page 6: "ATRIP is an activator of ATR". ATRIP is best described as a subunit of the ATR-ATRIP kinase complex (Cortez et al., Science 2001). The use of the word activator here could be confusing since the ATR signaling field uses this word to describe TOPBP1 and ETAA1 which bind and stimulate the ATR-ATRIP kinase complex.

3. The observation that partial reduction in ATR activity sensitizes to ATR inhibition was previously published (Mohani et al., Cancer Research 2014) and should be referenced on page 6 where the authors make a similar conclusion..

4. The authors do not do much follow-up of any of the screen hits within the manuscript except for asking if a histone deposition function of POLE3/POLE4 is involved in generating ATR inhibitor hyper-sensitivity. Their experiments suggest it is not. Developing a mechanistic understanding of why loss of these polymerase subunits causes ATR inhibitor hypersensitivity could be quite difficult (and beyond the scope of this manuscript). However, it is interesting that POLE and POLE2 were not identified in the screen. This is also true in the other CRISPR screen (Wang et al., Oncogene 2018). However, siRNA to POLE does cause hypersensitivity to ATR inhibitor in U2OS cells (Mohani et al., Plos One 2015 and Mohani et al., Cancer Research 2014). While it is possible that the siRNA experiments were caused by off-target effects, the simplest interpretation is that a reduction in DNA synthesis activity underlies the hypersensitivity. Did the authors check whether POLE or POLE2 loss of function either by CRISPR or RNA interference phenocopies POLE3/POLE4 ATR inhibitor hypersensitivity in their hands? Do they observe any effect of POLE3 and POLE4 inactivation on DNA synthesis rates? Even if no further experiments are completed, additional discussion about why these two subunits and not POLE or POLE2 are strong hits in their screens would be useful.

Decision letter (RSOB-19-0156.R0)

30-Jul-2019

Dear Dr Durocher,

We are pleased to inform you that your manuscript RSOB-19-0156 entitled "A consensus set of genetic vulnerabilities to ATR inhibition" has been accepted by the Editor for publication in Open Biology. The reviewer has recommended publication, but also suggest some minor revisions to your manuscript. Therefore, we invite you to respond to the reviewer's comments and revise your manuscript.

Please submit the revised version of your manuscript within 7 days. If you do not think you will be able to meet this date please let us know immediately and we can extend this deadline for you.

When submitting your revised manuscript, you will be able to respond to the comments made by

the referee(s) and upload a file "Response to Referees" in "Section 6 - File Upload". You can use this to document any changes you make to the original manuscript. In order to expedite the processing of the revised manuscript, please be as specific as possible in your response to the referee(s).

- 1) A text file of the manuscript (doc, txt, rtf or tex), including the references, tables (including captions) and figure captions. Please remove any tracked changes from the text before submission. PDF files are not an accepted format for the "Main Document".
- 2) A separate electronic file of each figure (tiff, EPS or print-quality PDF preferred). The format should be produced directly from original creation package, or original software format. Please note that PowerPoint files are not accepted.
- 3) Electronic supplementary material: this should be contained in a separate file from the main text and meet our ESM criteria (see <http://royalsocietypublishing.org/instructions-authors#question5>). All supplementary materials accompanying an accepted article will be treated as in their final form. They will be published alongside the paper on the journal website and posted on the online figshare repository. Files on figshare will be made available approximately one week before the accompanying article so that the supplementary material can be attributed a unique DOI.

Online supplementary material will also carry the title and description provided during submission, so please ensure these are accurate and informative. Note that the Royal Society will not edit or typeset supplementary material and it will be hosted as provided. Please ensure that the supplementary material includes the paper details (authors, title, journal name, article DOI). Your article DOI will be 10.1098/rsob.2016[*last 4 digits of e.g. 10.1098/rsob.20160049*].

- 4) A media summary: a short non-technical summary (up to 100 words) of the key findings/importance of your manuscript. Please try to write in simple English, avoid jargon, explain the importance of the topic, outline the main implications and describe why this topic is newsworthy.

Images

Data-Sharing

It is a condition of publication that data supporting your paper are made available. Data should be made available either in the electronic supplementary material or through an appropriate repository. Details of how to access data should be included in your paper. Please see <http://royalsocietypublishing.org/site/authors/policy.xhtml#question6> for more details.

Data accessibility section

- DNA sequences: Genbank accessions F234391-F234402
- Phylogenetic data: TreeBASE accession number S9123
- Final DNA sequence assembly uploaded as online supplemental material

- Climate data and MaxEnt input files: Dryad doi:10.5521/dryad.12311

Sincerely,

The Open Biology Team
mailto:openbiology@royalsociety.org

Reviewer's Comments to Author:

Referee:

Comments to the Author(s)

In this manuscript the authors report the results of CRISPR-Cas9 screens looking for genetic vulnerabilities to ATR inhibition. The results provide a useful resource for investigators in the DNA repair, replication, and DNA damage signaling fields. The writing and figures are clear and the conclusions are appropriate. I recommend publication. The following comments may assist the authors in improving the final version of the manuscript.

1. It is difficult to judge overlap between the 4 screens in the three cell types. I recommend including a simple Venn diagram.
2. Page 6: "ATRIP is an activator of ATR". ATRIP is best described as a subunit of the ATR-ATRIP kinase complex (Cortez et al., Science 2001). The use of the word activator here could be confusing since the ATR signaling field uses this word to describe TOPBP1 and ETAA1 which bind and stimulate the ATR-ATRIP kinase complex.
3. The observation that partial reduction in ATR activity sensitizes to ATR inhibition was previously published (Mohani et al., Cancer Research 2014) and should be referenced on page 6 where the authors make a similar conclusion..
4. The authors do not do much follow-up of any of the screen hits within the manuscript except for asking if a histone deposition function of POLE3/POLE4 is involved in generating ATR inhibitor hyper-sensitivity. Their experiments suggest it is not. Developing a mechanistic understanding of why loss of these polymerase subunits causes ATR inhibitor hypersensitivity could be quite difficult (and beyond the scope of this manuscript). However, it is interesting that POLE and POLE2 were not identified in the screen. This is also true in the other CRISPR screen (Wang et al., Oncogene 2018). However, siRNA to POLE does cause hypersensitivity to ATR inhibitor in U2OS cells (Mohani et al., Plos One 2015 and Mohani et al., Cancer Research 2014). While it is possible that the siRNA experiments were caused by off-target effects, the simplest interpretation is that a reduction in DNA synthesis activity underlies the hypersensitivity. Did the authors check whether POLE or POLE2 loss of function either by CRISPR or RNA interference phenocopies POLE3/POLE4 ATR inhibitor hypersensitivity in their hands? Do they observe any effect of POLE3 and POLE4 inactivation on DNA synthesis rates? Even if no further experiments are completed, additional discussion about why these two subunits and not POLE or POLE2 are strong hits in their screens would be useful.

Author's Response to Decision Letter for (RSOB-19-0156.R0)

See Appendix A.

Decision letter (RSOB-19-0156.R1)

22-Aug-2019

Dear Dr Durocher,

We are pleased to inform you that your manuscript entitled "A consensus set of genetic vulnerabilities to ATR inhibition" has been accepted by the Editor for publication in Open Biology.

Sincerely,

The Open Biology Team
mailto:openbiology@royalsociety.org

Appendix A

We sincerely thank the expert Reviewer for his/her comments. Below is our point-by-point response. The Reviewer's comments are italicized and a bullet point precedes our responses.

Referee #1:

In this manuscript the authors report the results of CRISPR-Cas9 screens looking for genetic vulnerabilities to ATR inhibition. The results provide a useful resource for investigators in the DNA repair, replication, and DNA damage signaling fields. The writing and figures are clear and the conclusions are appropriate. I recommend publication. The following comments may assist the authors in improving the final version of the manuscript.

1. *It is difficult to judge overlap between the 4 screens in the three cell types. I recommend including a simple Venn diagram.*
 - We thank the Reviewer for this comment. To help with screen comparison we generated an UpSet plot instead of a Venn diagram for the 7 screens. The two most related screens are the two screens done as part of this study in RPE1 cells. The UpSet plot is now in the revised Supplementary Fig 1c and is now referred in the text on p6.
2. *Page 6: "ATRIP is an activator of ATR". ATRIP is best described as a subunit of the ATR-ATRIP kinase complex (Cortez et al., Science 2001). The use of the word activator here could be confusing since the ATR signaling field uses this word to describe TOPBP1 and ETAA1 which bind and stimulate the ATR-ATRIP kinase complex.*
 - The Reviewer is correct and we have changed the text to state that ATRIP is a subunit of the ATR-ATRIP complex (see revised manuscript p6).
3. *The observation that partial reduction in ATR activity sensitizes to ATR inhibition was previously published (Mohani et al., Cancer Research 2014) and should be referenced on page 6 where the authors make a similar conclusion.*
 - Thank you for noting our omission. The reference to Mohani et al. 2014 is now included in the revised manuscript (p6).
4. *The authors do not do much follow-up of any of the screen hits within the manuscript except for asking if a histone deposition function of POLE3/POLE4 is involved in generating ATR inhibitor hyper-sensitivity. Their experiments suggest it is not. Developing a mechanistic understanding of why loss of these polymerase subunits causes ATR inhibitor hypersensitivity could be quite difficult (and beyond the scope of this manuscript). However, it is interesting that POLE and POLE2 were not identified in the screen. This is also true in the other CRISPR screen (Wang et al., Oncogene 2018). However, siRNA to POLE does cause hypersensitivity to ATR inhibitor in U2OS cells (Mohani et al., Plos One 2015 and Mohani et al., Cancer Research 2014). While it is*

possible that the siRNA experiments were caused by off-target effects, the simplest interpretation is that a reduction in DNA synthesis activity underlies the hypersensitivity. Did the authors check whether POLE or POLE2 loss of function either by CRISPR or RNA interference phenocopies POLE3/POLE4 ATR inhibitor hypersensitivity in their hands? Do they observe any effect of POLE3 and POLE4 inactivation on DNA synthesis rates? Even if no further experiments are completed, additional discussion about why these two subunits and not POLE or POLE2 are strong hits in their screens would be useful.

- The Reviewer is correct in raising the potential role for POLE and POLE2 in mediating resistance to ATR inhibitors. Challenged DNA replication is a well-known vulnerability to ATR inhibition. However, it is very difficult to comment on negative results in genetic screens, and this is especially relevant for genes that are essential. It may well be that the depletion of the sgRNAs against *POLE* and *POLE2* was too severe in our edited populations to get a reliable quantitation prior and post ATRi treatment. In our opinion, this remains the likeliest explanation.

That being said, while null *Pole4*^{-/-} mouse cells show evidence of DNA replication stress, Bellelli et al. (Bellelli et al., 2018) recently analysed the phenotypes of *POLE3/POLE4* depletion by siRNA and the authors concluded that transient depletion of *POLE3/POLE4* does “not induce significant levels of replication stress nor obvious changes in cell cycle profile or BrdU incorporation”. This is consistent with the observation that *POLE3/4* are dispensable for in vitro DNA replication (Goswami et al., 2018). Therefore, while we now acknowledge that perturbation in Pol e activity may be the root cause of the hypersensitivity of *POLE3/4*-deficient cells to ATRi (p8), it remains a distinct possibility that the *POLE3/POLE4* accessory subunits coordinate a replication-associated process, distinct from DNA polymerisation, that allows cells to survive upon ATR inhibitor treatment. We have modified the manuscript to briefly discuss these points (see revised manuscript p8).

References

- Bellelli, R., Belan, O., Pye, V.E., Clement, C., Maslen, S.L., Skehel, J.M., Cherepanov, P., Almouzni, G., and Boulton, S.J. (2018). POLE3-POLE4 Is a Histone H3-H4 Chaperone that Maintains Chromatin Integrity during DNA Replication. *Mol Cell* 72, 112-126 e115.
- Goswami, P., Abid Ali, F., Douglas, M.E., Locke, J., Purkiss, A., Janska, A., Eickhoff, P., Early, A., Nans, A., Cheung, A.M.C., et al. (2018). Structure of DNA-CMG-Pol epsilon elucidates the roles of the non-catalytic polymerase modules in the eukaryotic replisome. *Nature communications* 9, 5061.